# Universal Invariant and Equivariant Graph Neural Networks

**Nicolas Keriven**
École Normale Supérieure
Paris, France
nicolas.keriven@ens.fr

**Gabriel Peyré**
CNRS and École Normale Supérieure
Paris, France
gabriel.peyre@ens.fr

## Abstract

Graph Neural Networks (GNN) come in many flavors, but should always be either *invariant* (permutation of the nodes of the input graph does not affect the output) or *equivariant* (permutation of the input permutes the output). In this paper, we consider a specific class of invariant and equivariant networks, for which we prove new universality theorems. More precisely, we consider networks with a single hidden layer, obtained by summing channels formed by applying an equivariant linear operator, a pointwise non-linearity, and either an invariant or equivariant linear output layer. Recently, Maron et al. (2019b) showed that by allowing higher-order tensorization inside the network, universal *invariant* GNNs can be obtained. As a first contribution, we propose an alternative proof of this result, which relies on the Stone-Weierstrass theorem for algebra of real-valued functions. Our main contribution is then an extension of this result to the *equivariant* case, which appears in many practical applications but has been less studied from a theoretical point of view. The proof relies on a new generalized Stone-Weierstrass theorem for algebra of equivariant functions, which is of independent interest. Additionally, unlike many previous works that consider a fixed number of nodes, our results show that a GNN defined by a single set of parameters can approximate uniformly well a function defined on graphs of varying size.

## 1 Introduction

Designing Neural Networks (NN) to exhibit some *invariance* or *equivariance* to group operations is a central problem in machine learning (Shawe-Taylor, 1993). Among these, Graph Neural Networks (GNN) are primary examples that have gathered a lot of attention for a large range of applications. Indeed, since a graph is not changed by permutation of its nodes, GNNs must be either *invariant to permutation*, if they return a result that must not depend on the representation of the input, or *equivariant to permutation*, if the output must be permuted when the input is permuted, for instance when the network returns a *signal over the nodes* of the input graph. In this paper, we examine universal approximation theorems for invariant and equivariant GNNs.

From a theoretical point of view, invariant GNNs have been much more studied than their equivariant counterpart (see the following subsection). However, many practical applications deal with equivariance instead, such as community detection (Chen et al., 2019), recommender systems (Ying et al., 2018), interaction networks of physical systems (Battaglia et al., 2016), state prediction (Sanchez-Gonzalez et al., 2018), protein interface prediction (Fout et al., 2017), among many others. See (Zhou et al., 2018; Bronstein et al., 2017) for thorough reviews. It is therefore of great interest to increase our understanding of equivariant networks, in particular, by extending arguably one of the most classical result on neural networks, namely the universal approximation theorem for multi-layers perceptron (MLP) with a single hidden layer (Cybenko, 1989; Hornik et al., 1989; Pinkus, 1999).

Maron et al. (2019b) recently proved that certain *invariant* GNNs were universal approximators of invariant continuous functions on graphs. The main goal of this paper is to extend this result to the *equivariant* case, for similar architectures.

**Outline and contribution.** The outline of our paper is as follows. After reviewing previous works and notations in the rest of the introduction, in Section 2 we provide an alternative proof of the result of (Maron et al., 2019b) for invariant GNNs (Theorem 1), which will serve as a basis for the equivariant case. It relies on a non-trivial application of the classical Stone-Weierstrass theorem for algebras of real-valued functions (recalled in Theorem 2). Then, as our main contribution, in Section 3 we prove this result for the equivariant case (Theorem 3), which to the best of our knowledge was not known before. The proof relies on a new version of Stone-Weierstrass theorem (Theorem 4). Unlike many works that consider a fixed number of nodes $n$, in both cases we will prove that a GNN described by a single set of parameters can approximate uniformly well a function that acts on graphs of varying size.

## 1.1 Previous works

The design of neural network architectures which are equivariant or invariant under group actions is an active area of research, see for instance (Ravanbakhsh et al., 2017; Gens and Domingos, 2014; Cohen and Welling, 2016) for finite groups and (Wood and Shawe-Taylor, 1996; Kondor and Trivedi, 2018) for infinite groups. We focus here our attention to discrete groups acting on the coordinates of the features, and more specifically to the action of the full set of permutations on tensors (order-1 tensors corresponding to sets, order-2 to graphs, order-3 to triangulations, etc).

**Convolutional GNN.** The most appealing construction of GNN architectures is through the use of local operators acting on vectors indexed by the vertices. Early definitions of these "message passing" architectures rely on fixed point iterations (Scarselli et al., 2009), while more recent constructions make use of non-linear functions of the adjacency matrix, for instance using spectral decompositions (Bruna et al., 2014) or polynomials (Defferrard et al., 2016). We refer to (Bronstein et al., 2017; Xu et al., 2019) for recent reviews. For regular-grid graphs, they match classical convolutional networks (LeCun et al., 1989) which by design can only approximate translation-invariant or equivariant functions (Yarotsky, 2018). It thus comes at no surprise that these convolutional GNN are not universal approximators (Xu et al., 2019) of permutation-invariant functions.

**Fully-invariant GNN.** Designing Graph (and their higher-dimensional generalizations) NN which are equivariant or invariant to the whole permutation group (as opposed to e.g. only translations) requires the use of a small sub-space of linear operators, which is identified in (Maron et al., 2019a). This generalizes several previous constructions, for instance for sets (Zaheer et al., 2017; Hartford et al., 2018) and points clouds (Qi et al., 2017). Universality results are known to hold in the special cases of sets, point clouds (Qi et al., 2017) and discrete measures (de Bie et al., 2019) networks.

In the *invariant* GNN case, the universality of architectures built using a single hidden layer of such equivariant operators followed by an invariant layer is proved in (Maron et al., 2019b) (see also (Kondor et al., 2018)). This is the closest work from our, and we will provide an alternative proof of this result in Section 2, as a basis for our main result in Section 3.

Universality in the equivariant case has been less studied. Most of the literature focuses on equivariance to *translation* and its relation to convolutions (Kondor et al., 2018; Cohen and Welling, 2016), which are ubiquitous in image processing. In this context, Yarotsky (2018) proved the universality of some translation-equivariant networks. Closer to our work, universality of NNs equivariant to permutations acting on point clouds has been recently proven in (Sannai et al., 2019), however their theorem does not allow for high-order inputs like graphs. It is the purpose of our paper to fill this missing piece and prove the universality of a class of equivariant GNNs for high-order inputs such as (hyper-)graphs.

## 1.2 Notations and definitions

**Graphs.** In this paper, (hyper-)graphs with $n$ nodes are represented by tensors $G \in \mathbb{R}^{n^d}$ indexed by $1 \leqslant i_1, \ldots, i_d \leqslant n$. For instance, "classical" graphs are represented by edge weight matrices ($d = 2$), and hyper-graphs by high-order tensors of "multi-edges" connecting more than two nodes.

Note that we do not impose $G$ to be symmetric, or to contain only non-negative elements. In the rest of the paper, we fix some $d \geqslant 1$ for the order of the inputs, however we allow $n$ to vary.

**Permutations.** Let $[n] \overset{\text{def.}}{=} \{1, \ldots, n\}$. The set of permutations $\sigma : [n] \to [n]$ (bijections from $[n]$ to itself) is denoted by $\mathcal{O}_n$, or simply $\mathcal{O}$ when there is no ambiguity. Given a permutation $\sigma$ and an order-$k$ tensor $G \in \mathbb{R}^{n^k}$, a "permutation of nodes" on $G$ is denoted by $\sigma \star G$ and defined as

$$(\sigma \star G)_{\sigma(i_1), \ldots, \sigma(i_k)} = G_{i_1, \ldots, i_k}.$$

We denote by $P_\sigma \in \{0, 1\}^{n \times n}$ the permutation matrix corresponding to $\sigma$, or simply $P$ when there is no ambiguity. For instance, for $G \in \mathbb{R}^{n^2}$ we have $\sigma \star G = PGP^\top$.

Two graphs $G_1, G_2$ are said isomorphic if there is a permutation $\sigma$ such that $G_1 = \sigma \star G_2$. If $G = \sigma \star G$, we say that $\sigma$ is a self-isomorphism of $G$. Finally, we denote by $\mathcal{O}(G) \overset{\text{def.}}{=} \{\sigma \star G ;\ \sigma \in \mathcal{O}\}$ the orbit of all the permuted versions of $G$.

**Invariant and equivariant linear operators.** A function $f : \mathbb{R}^{n^k} \to \mathbb{R}$ is said to be *invariant* if $f(\sigma \star G) = f(G)$ for every permutation $\sigma$. A function $f : \mathbb{R}^{n^k} \to \mathbb{R}^{n^\ell}$ is said to be *equivariant* if $f(\sigma \star G) = \sigma \star f(G)$. Our construction of GNNs alternates between *linear* operators that are invariant or equivariant to permutations, and non-linearities. Maron et al. (2019a) elegantly characterize all such linear functions, and prove that they live in vector spaces of dimension, respectively, exactly $b(k)$ and $b(k + \ell)$, where $b(i)$ is the $i^{th}$ Bell number. An important corollary of this result is that the dimension of this space *does not depend on the number of nodes* $n$, but only on the order of the input and output tensors. Therefore one can parameterize linearly for all $n$ such an operator by the same set of coefficients. For instance, a linear equivariant operator $F : \mathbb{R}^{n^2} \to \mathbb{R}^{n^2}$ from matrices to matrices is formed by a linear combination of $b(4) = 15$ basic operators such as "sum of rows replicated on the diagonal", "sum of columns replicated on the rows", and so on. The 15 coefficients used in this linear combination define the "same" linear operator for every $n$.

**Invariant and equivariant Graph Neural Nets.** As noted by Yarotsky (2018), it is in fact trivial to build invariant universal networks for finite groups of symmetry: just take a non-invariant universal architecture, and perform a group averaging. However, this holds little interest in practice, since the group of permutation is of size $n!$. Instead, researchers use architectures for which invariance is hard-coded into the construction of the network itself. The same remark holds for equivariance.

In this paper, we consider one-layer GNNs of the form:

$$f(G) = \sum_{s=1}^{S} H_s \Big[ \rho(F_s[G] + B_s) \Big] + b, \tag{1}$$

where $F_s : \mathbb{R}^{n^d} \to \mathbb{R}^{n^{k_s}}$ are linear equivariant functions that yield $k_s$-tensors (i.e. they *potentially increase or decrease the order of the input tensor*), and $H_s$ are invariant linear operators $H_s : \mathbb{R}^{n^{k_s}} \to \mathbb{R}$ (resp. equivariant linear operators $H_s : \mathbb{R}^{n^{k_s}} \to \mathbb{R}^n$), such that the GNN is globally invariant (resp. equivariant). The invariant case is studied in Section 2, and the equivariant in Section 3. The bias terms $B_s \in \mathbb{R}^{n^{k_s}}$ are equivariant, so that $B_s = \sigma \star B_s$ for all $\sigma$. They are also characterized by Maron et al. (2019a) and belong to a linear space of dimension $b(k_s)$. We illustrate this simple architecture in Fig. 1.

In light of the characterization by Maron et al. (2019a) of linear invariant and equivariant operators described in the previous paragraph, a GNN of the form (1) is described by $1 + \sum_{s=1}^{S} b(k_s + d) + 2b(k_s)$ parameters in the invariant case and $1 + \sum_{s=1}^{S} b(k_s + d) + b(k_s + 1) + b(k_s)$ in the equivariant. As mentioned earlier, this number of parameters does not depend on the number of nodes $n$, and a GNN described by a single set of parameters can be applied to graphs of any size. In particular, we are going to show that a GNN approximates uniformly well a continuous function for several $n$ at once.

The function $\rho$ is any locally Lipschitz pointwise non-linearity for which the Universal Approximation Theorem for MLP applies. We denote their set $\mathcal{F}_{\mathrm{MLP}}$. This includes in particular any continuous function that is not a polynomial (Pinkus, 1999). Among these, we denote the sigmoid $\rho_{\mathrm{sig}}(x) = e^x/(1 + e^x)$.

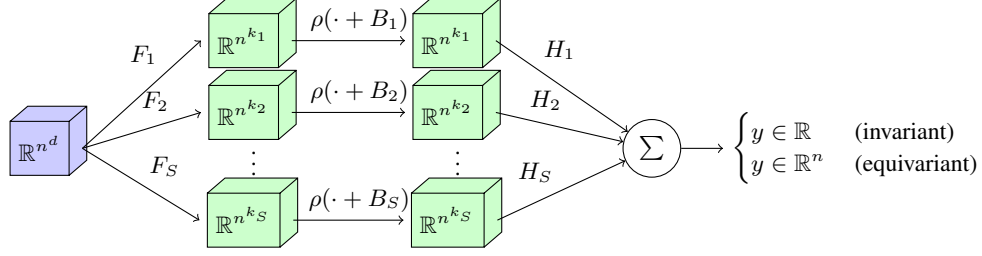

Figure 1: The model of GNNs studied in this paper. For each channel $s \leqslant S$, the input tensor is passed through an equivariant operator $F_s : \mathbb{R}^{n^d} \to \mathbb{R}^{n^{k_s}}$, a non-linearity with some added equivariant bias $B_s$, and a final operator $H_s$ that is either invariant (Section 2) or equivariant (Section 3). These GNNs are universal approximators of invariant or equivariant continuous functions (Theorems 1 and 3).

We denote by $\mathcal{N}_{\text{inv.}}(\rho)$ (resp. $\mathcal{N}_{\text{eq.}}(\rho)$) the class of invariant (resp. equivariant) 1-layer networks of the form (1) (with $S$ and $k_s$ being arbitrarily large). Our contributions show that they are dense in the spaces of continuous invariant (resp. equivariant) functions.

## 2   The case of invariant functions

Maron et al. (2019b) recently proved that *invariant* GNNs similar to (1) are universal approximators of continuous invariant functions. As a warm-up, we propose an alternative proof of (a variant of) this result, that will serve as a basis for our main contribution, the equivariant case (Section 3).

**Edit distance.**   For invariant functions, isomorphic graphs are undistinguishable, and therefore we work with a set of *equivalence classes* of graphs, where two graphs are equivalent if isomorphic. We define such a set for any number $n \leqslant n_{\max}$ of nodes and bounded $G$

$$\mathcal{G}_{\text{inv.}} \stackrel{\text{def.}}{=} \left\{ \mathcal{O}(G) \; ; \; G \in \mathbb{R}^{n^d} \text{ with } n \leqslant n_{\max}, \|G\| \leqslant R \right\},$$

where we recall that $\mathcal{O}(G) = \{\sigma \star G \; ; \; \sigma \in \mathcal{O}\}$ is the set of every permuted versions of $G$, here seen as an equivalence class.

We need to equip this set with a metric that takes into account graphs with different number of nodes. A distance often used in the literature is the *graph edit distance* (Sanfeliu and Fu, 1983). It relies on defining a set of elementary operations $o$ and a cost $c(o)$ associated to each of them, here we consider node addition and edge weight modification. The distance is then defined as

$$d_{\text{edit}}(\mathcal{O}(G_1), \mathcal{O}(G_2)) \stackrel{\text{def.}}{=} \min_{(o_1, \ldots, o_k) \in \mathcal{P}(G_1, G_2)} \sum_{i=1}^{k} c(o_i) \tag{2}$$

where $\mathcal{P}(G_1, G_2)$ contains every sequence of operation to transform $G_1$ into a graph isomorphic to $G_2$, or $G_2$ into $G_1$. Here we consider $c(node\_addition) = c$ for some constant $c > 0$, $c(edge\_weight\_change) = |w - w'|$ where the weight change is from $w$ to $w'$, and "edge" refers to any element of the tensor $G \in \mathbb{R}^{n^d}$. Note that, if we have $d_{\text{edit}}(\mathcal{O}(G_1), \mathcal{O}(G_2)) < c$, then $G_1$ and $G_2$ have the same number of nodes, and in that case $d_{\text{edit}}(\mathcal{O}(G_1), \mathcal{O}(G_2)) = \min_{\sigma \in \mathcal{O}_n} \|G_1 - \sigma \star G_2\|_1$, where $\|\cdot\|_1$ is the element-wise $\ell_1$ norm, since each edge must be transformed into another.

We denote by $\mathcal{C}(\mathcal{G}_{\text{inv.}}, d_{\text{edit}})$ the space of real-valued functions on $\mathcal{G}_{\text{inv.}}$ that are continuous with respect to $d_{\text{edit}}$, equipped with the infinity norm of uniform convergence. We then have the following result.

**Theorem 1.** *For any $\rho \in \mathcal{F}_{\text{MLP}}$, $\mathcal{N}_{\text{inv.}}(\rho)$ is dense in $\mathcal{C}(\mathcal{G}_{\text{inv.}}, d_{edit})$.*

**Comparison with (Maron et al., 2019b).**   A variant of Theorem 1 was proved in (Maron et al., 2019b). The two proofs are however different: their proof relies on the construction of a basis of invariant polynomials and on classical universality of MLPs, while our proof is a direct application of Stone-Weierstrass theorem for algebras of real-valued functions. See the next subsection for details.

One improvement of our result with respect to the one of (Maron et al., 2019b) is that it can handle graphs of varying sizes. As mentioned in the introduction, a single set of parameters defines a GNN that can be applied to graphs of any size. Theorem 1 shows that any continuous invariant function is *uniformly* well approximated by a GNN on the whole set $\mathcal{G}_{\text{inv.}}$, that is, for all numbers of nodes $n \leqslant n_{\max}$ simultaneously. On the contrary, Maron et al. (2019b) work with a fixed $n$, and it does not seem that their proof can extend easily to encompass several $n$ at once. A weakness of our proof is that it does not provide an upper bound on the order of tensorization $k_s$. Indeed, through Noether's theorem on polynomials, the proof of Maron et al. (2019b) shows that $k_s \leqslant n^d(n^d - 1)/2$ is sufficient for universality, which we cannot seem to deduce from our proof. Moreover, they provide a lower-bound $k_s \geqslant n^d$ below which universality cannot be achieved.

## 2.1 Sketch of proof of Theorem 1

The proof for the invariant case will serve as a basis for the equivariant case in the Section 3. It relies on Stone-Weierstrass theorem, which we recall below.

**Theorem 2** (Stone-Weierstrass (Rudin (1991), Thm. 5.7)). *Suppose $X$ is a compact Hausdorff space and $A$ is a subalgebra of the space of continuous real-valued functions $\mathcal{C}(X)$ which contains a non-zero constant function. Then $\mathcal{A}$ is dense in $\mathcal{C}(X)$ if and only if it separates points, that is, for all $x \neq y$ in $X$ there exists $f \in \mathcal{A}$ such that $f(x) \neq f(y)$.*

We will construct a class of GNNs that satisfy all these properties in $\mathcal{G}_{\text{inv.}}$. As we will see, unlike classical applications of this theorem to e.g. polynomials, the main difficulty here will be to prove the separation of points. We start by observing that $\mathcal{G}_{\text{inv.}}$ is indeed a compact set for $d_{\text{edit}}$.

**Properties of $(\mathcal{G}_{\text{inv.}}, d_{\text{edit}})$.** Let us first note that the metric space $(\mathcal{G}_{\text{inv.}}, d_{\text{edit}})$ is Hausdorff (i.e. separable, all metric spaces are). For each $\mathcal{O}(G_1), \mathcal{O}(G_2) \in \mathcal{G}_{\text{inv.}}$ we have: if $d_{\text{edit}}(\mathcal{O}(G_1), \mathcal{O}(G_2)) < c$, then the graphs have the same number of nodes, and in that case $d_{\text{edit}}(\mathcal{O}(G_1)\mathcal{O}(G_2)) \leqslant \|G_1 - G_2\|_1$. Therefore, the embedding $G \mapsto \mathcal{O}(G)$ is continuous (locally Lipschitz). As the continuous image of the compact $\bigcup_{n=1}^{n_{\max}} \left\{ G \in \mathbb{R}^{n^d} \; ; \; \|G\| \leqslant R \right\}$, the set $\mathcal{G}_{\text{inv.}}$ is indeed compact.

**Algebra of invariant GNNs.** Unfortunately, $\mathcal{N}_{\text{inv.}}(\rho)$ is not a subalgebra. Following Hornik et al. (1989), we first need to extend it to be closed under multiplication. We do that by allowing Kronecker products inside the invariant functions:

$$f(G) = \sum_{s=1}^{S} H_s \Big[ \rho\left(F_{s1}[G] + B_{s1}\right) \otimes \ldots \otimes \rho\left(F_{sT_s}[G] + B_{sT_s}\right) \Big] + b \tag{3}$$

where $F_{st}$ yields $k_{st}$-tensors, $H_s : \mathbb{R}^{n^{\sum_t k_{st}}} \to \mathbb{R}$ are invariant, and $B_{st}$ are equivariant bias. By $(\sigma \star G) \otimes (\sigma \star G') = \sigma \star (G \otimes G')$, they are indeed invariant. We denote by $\mathcal{N}_{\text{inv.}}^{\otimes}(\rho)$ the set of all GNNs of this form, with $S, T_s, k_{st}$ arbitrarily large.

**Lemma 1.** *For any locally Lipschitz $\rho$, $\mathcal{N}_{\text{inv.}}^{\otimes}(\rho)$ is a subalgebra in $\mathcal{C}(\mathcal{G}_{\text{inv.}}, d_{edit})$.*

The proof, presented in Appendix A.1.1 follows from manipulations of Kronecker products.

**Separability.** The main difficulty in applying Stone-Weierstrass theorem is the separation of points, which we prove in the next Lemma.

**Lemma 2.** *$\mathcal{N}_{\text{inv.}}^{\otimes}(\rho_{sig})$ separates points.*

The proof, presented in Appendix A.1.2, proceeds by contradiction: we show that two graphs $G, G'$ that coincides for every GNNs are necessarily permutation of each other. Applying Stone-Weierstrass theorem, we have thus proved that $\mathcal{N}_{\text{inv.}}^{\otimes}(\rho_{\text{sig}})$ is dense in $\mathcal{C}(\mathcal{G}_{\text{inv.}}, d_{\text{edit}})$.

Then, following Hornik et al. (1989), we go back to the original class $\mathcal{N}_{\text{inv.}}(\rho)$, by applying: $(i)$ a Fourier approximation of $\rho_{\text{sig}}$, $(ii)$ the fact that a product of cos is also a sum of cos, and $(iii)$ an approximation of cos by any other non-linearity. The following Lemma is proved in Appendix A.1.3, and concludes the proof of Thm 1.

**Lemma 3.** *We have the following: $(i)$ $\mathcal{N}_{\text{inv.}}^{\otimes}(\cos)$ is dense in $\mathcal{N}_{\text{inv.}}^{\otimes}(\rho_{sig})$; $(ii)$ $\mathcal{N}_{\text{inv.}}^{\otimes}(\cos) = \mathcal{N}_{\text{inv.}}(\cos)$; $(iii)$ for any $\rho \in \mathcal{F}_{\text{MLP}}$, $\mathcal{N}_{\text{inv.}}(\rho)$ is dense in $\mathcal{N}_{\text{inv.}}(\cos)$.*

# 3 The case of equivariant functions

This section contains our main contribution. We examine the case of equivariant functions that return a vector $f(G) \in \mathbb{R}^n$ when $G$ has $n$ nodes, such that $f(\sigma \star G) = \sigma \star f(G)$. In that case, isomorphic graphs are not equivalent anymore. Hence we consider a compact set of graphs

$$\mathcal{G}_{\text{eq.}} \stackrel{\text{def.}}{=} \left\{ G \in \mathbb{R}^{n^d} \; ; \; n \leqslant n_{\max}, \|G\| \leqslant R \right\},$$

Like the invariant case, we consider several numbers of nodes $n \leqslant n_{\max}$ and will prove uniform approximation over them. We do not use the edit distance but a simpler metric:

$$d(G, G') = \begin{cases} \|G - G'\| & \text{if } G \text{ and } G' \text{ have the same number of nodes,} \\ \infty & \text{otherwise.} \end{cases}$$

for any norm $\|\cdot\|$ on $\mathbb{R}^{n^d}$.

The set of equivariant continuous functions is denoted by $\mathcal{C}_{\text{eq.}}(\mathcal{G}_{\text{eq.}}, d)$, equipped with the infinity norm $\|f\|_\infty = \sup_{G \in \mathcal{G}_{\text{eq.}}} \|f(G)\|_\infty$. We recall that $\mathcal{N}_{\text{eq.}}(\rho) \subset \mathcal{C}_{\text{eq.}}(\mathcal{G}_{\text{eq.}}, d)$ denotes one-layer GNNs of the form (1), with equivariant output operators $H_s$. Our main result is the following.

**Theorem 3.** *For any $\rho \in \mathcal{F}_{\text{MLP}}$, $\mathcal{N}_{\text{eq.}}(\rho)$ is dense in $\mathcal{C}_{\text{eq.}}(\mathcal{G}_{\text{eq.}}, d)$.*

The proof, detailed in the next section, follows closely the previous proof for invariant functions, but is significantly more involved. Indeed, the classical version of Stone-Weierstrass only provides density of a subalgebra of functions in the *whole space* of continuous functions, while in this case $\mathcal{C}_{\text{eq.}}(\mathcal{G}_{\text{eq.}}, d)$ is *already* a particular subset of continuous functions. On the other hand, it seems difficult to make use of fully general versions of Stone-Weierstrass theorem, for which some questions are still open (Glimm, 1960). Hence we prove a new, specialized Stone-Weierstrass theorem for equivariant functions (Theorem 4), obtained with a non-trivial adaptation of the constructive proof by Brosowski and Deutsch (1981).

Like the invariant case, our theorem proves uniform approximation for all numbers of nodes $n \leqslant n_{\max}$ at once by a single GNN. As is detailed in the next subsection, our proof of the generalized Stone-Weierstrass theorem relies on being able to *sort* the coordinates of the output space $\mathbb{R}^n$, and therefore our current proof technique does not extend to high-order *output* $\mathbb{R}^{n^\ell}$ (graph to graph mappings), which we leave for future work. For the same reason, while the previous invariant case could be easily extended to invariance to *subgroups* of $\mathcal{O}_n$, as is done by Maron et al. (2019b), for the equivariant case our theorem only applies when considering the full permuation group $\mathcal{O}_n$. Nevertheless, our generalized Stone-Weierstrass theorem may be applicable in other contexts where equivariance to permutation is a desirable property.

**Comparison with (Sannai et al., 2019).** Sannai et al. (2019) recently proved that equivariant NNs acting *on point clouds* are universal, that is, for $d = 1$ in our notations. Despite the apparent similarity with our result, there is a fundamental obstruction to extending their proof to high-order input tensors like graphs. Indeed, it strongly relies on Theorem 2 of (Zaheer et al., 2017) that characterizes invariant functions $\mathbb{R}^n \to \mathbb{R}$, which is no longer valid for high-order inputs.

## 3.1 Sketch of proof of Theorem 3: an equivariant version of Stone-Weierstrass theorem

We first need to introduce a few more notations. For a subset $I \subset [n]$, we define $\mathcal{O}_I \stackrel{\text{def.}}{=} \{\sigma \in \mathcal{O}_n \; ; \; \exists i \in I, j \in I^c, \sigma(i) = j \text{ or } \sigma(j) = i\}$ the set of permutations that exchange at least one index between $I$ and $I^c$. Indexing of vectors (or multivariate functions) is denoted by brackets, e.g. $[x]_I$ or $[f]_I$, and inequalities $x \geqslant a$ are to be understood element-wise.

**A new Stone-Weierstrass theorem.** We define the "multiplication" of two multivariate functions using the Hadamard product $\odot$, i.e. the component-wise multiplication. Since $(\sigma \star x) \odot (\sigma \star x') = \sigma \star (x \odot x')$, it is easy to see that $\mathcal{C}_{\text{eq.}}(\mathcal{G}_{\text{eq.}}, d)$ is closed under multiplication, and is therefore a (strict) subalgebra of the set of all continuous functions that return a vector in $\mathbb{R}^n$ for an input graph with $n$ nodes. As mentioned before, because of this last fact we cannot directly apply Stone-Weierstrass theorem. We therefore prove a new generalized version.

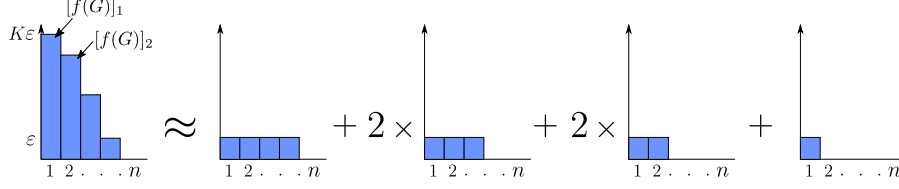

Figure 2: Illustration of strategy of proof for the equivariant Stone-Weierstrass theorem (Theorem 4). Considering a function $f$ that we are trying to approximate and a graph $G$ for which the coordinates of $f(G)$ are sorted by decreasing order, we approximate $f(G)$ by summing step-functions $f_i$, whose first coordinates are close to 1, and otherwise close to 0.

**Theorem 4** (Stone-Weierstrass for equivariant functions). *Let $\mathcal{A}$ be a subalgebra of $\mathcal{C}_{\text{eq.}}(\mathcal{G}_{\text{eq.}}, d)$, such that $\mathcal{A}$ contains the constant function $\mathbf{1}$ and:*

– *(Separability) for all $G, G' \in \mathcal{G}_{\text{eq.}}$ with number of nodes respectively $n$ and $n'$ such that $G \notin \mathcal{O}(G')$, for any $k \in [n]$, $k' \in [n']$, there exists $f \in \mathcal{A}$ such that $[f(G)]_k \neq [f(G')]_{k'}$ ;*

– *("Self"-separability) for all number of nodes $n \leqslant n_{\max}$, $I \subset [n]$, $G \in \mathcal{G}_{\text{eq.}}$ with $n$ nodes that has no self-isomorphism in $\mathcal{O}_I$, and $k \in I$, $\ell \in I^c$, there is $f \in \mathcal{A}$ such that $[f(G)]_k \neq [f(G)]_\ell$.*

*Then $\mathcal{A}$ is dense in $\mathcal{C}_{\text{eq.}}(\mathcal{G}_{\text{eq.}}, d)$.*

In addition to a "separability" hypothesis, which is similar to the classical one, Theorem 4 requires a "self"-separability condition, which guarantees that $f(G)$ can have different values on its coordinates under appropriate assumptions on $G$. We give below an overview of the proof of Theorem 4, the full details can be found in Appendix B.

Our proof is inspired by the one for the classical Stone-Weierstrass theorem (Thm. 2) of Brosowski and Deutsch (1981). Let us first give a bit of intuition on this earlier proof. It relies on the explicit construction of "step"-functions: given two disjoint closed sets $A$ and $B$, they show that $\mathcal{A}$ contains functions that are approximately 0 on $A$ and approximately 1 on $B$. Then, given a function $f : X \to \mathbb{R}$ (non-negative w.l.o.g.) that we are trying to approximate and $\varepsilon > 0$, they define $A_k = \{x \,;\, f(x) \leqslant (k - 1/3)\varepsilon\}$ and $B_k = \{x \,;\, f(x) \geqslant (k + 1/3)\varepsilon\}$ as the lower (resp. upper) *level sets* of $f$ for a grid of values with precision $\varepsilon$. Then, taking the step-functions $f_k$ between $A_k$ and $B_k$, it is easy to prove that $f$ is well-approximated by $g = \varepsilon \sum_k f_k$, since for each $x$ only the right number of $f_k$ is close to 1, the others are close to 0.

The situation is more complicated in our case. Given a function $f \in \mathcal{C}_{\text{eq.}}(\mathcal{G}_{\text{eq.}}, d)$ that we want to approximate, we work in the compact subset of $\mathcal{G}_{\text{eq.}}$ where the coordinates of $f$ are *ordered*, since by permutation it covers every case: $\mathcal{G}_f \stackrel{\text{def.}}{=} \left\{ G \in \mathcal{G}_{\text{eq.}} \,;\, \text{if } G \in \mathbb{R}^{n^d} : [f(G)]_1 \geqslant [f(G)]_2 \geqslant \ldots \geqslant [f(G)]_n \right\}$. Then, we will prove the existence of step-functions such that: when $A$ and $B$ satisfy some appropriate hypotheses, the step-function is close to 0 on $A$, and *only the first coordinates are close to* 1 on $B$, the others are close to 0. Indeed, by combining such functions, we can approximate a vector of ordered coordinates (Fig. 2). The construction of such step-functions is done in Lemma 7. Finally, we consider modified level-sets

$$A_k^{n,\ell} \stackrel{\text{def.}}{=} \left\{ G \in \mathcal{G}_f \cap \mathbb{R}^{n^d} \,;\, [f(G)]_\ell - [f(G)]_{\ell+1} \leqslant (k - 1/3)\varepsilon \right\} \cup \bigcup_{n' \neq n} \left( \mathcal{G}_f \cap \mathbb{R}^{(n')^d} \right),$$

$$B_k^{n,\ell} \stackrel{\text{def.}}{=} \left\{ G \in \mathcal{G}_f \cap \mathbb{R}^{n^d} \,;\, [f(G)]_\ell - [f(G)]_{\ell+1} \geqslant (k + 1/3)\varepsilon \right\}$$

that distinguish "jumps" between (ordered) coordinates. We define the associated step-functions $f_k^{n,\ell}$, and show that $g = \varepsilon \sum_{k,n,\ell} f_k^{n,\ell}$ is a valid approximation of $f$.

**End of the proof.** The rest of the proof of Theorem 3 is similar to the invariant case. We first build an algebra of GNNs, again by considering nets of the form (3), where we replace the $H_s$'s by equivariant linear operators in this case. We denote this space by $\mathcal{N}_{\text{eq.}}^{\otimes}(\rho)$.

**Lemma 4.** $\mathcal{N}_{\text{eq.}}^{\otimes}(\rho)$ *is a subalgebra of $\mathcal{C}_{\text{eq.}}(\mathcal{G}_{\text{eq.}}, d)$.*

The proof, presented in Appendix A.2.1, is very similar to that of Lemma 1. Then we show the two separation conditions for equivariant GNNs.

**Lemma 5.** $\mathcal{N}_{\text{eq.}}^{\otimes}(\rho_{sig.})$ *satisfies both the separability and self-separability conditions.*

The proof is presented in Appendix A.2.2. The "normal" separability is in fact equivalent to the previous one (Lemma 2), since we can construct an equivariant network by simply stacking an invariant network on every coordinate. The self-separability condition is proved in a similar way. Finally we go back to $\mathcal{N}_{\text{eq.}}(\rho)$ in exactly the same way. The proof of Lemma 6 is exactly similar to that of Lemma 3 and is omitted.

**Lemma 6.** *We have the following:* $(i)$ $\mathcal{N}_{\text{eq.}}^{\otimes}(\cos)$ *is dense in* $\mathcal{N}_{\text{eq.}}^{\otimes}(\rho_{sig})$; $(ii)$ $\mathcal{N}_{\text{eq.}}^{\otimes}(\cos) = \mathcal{N}_{\text{eq.}}(\cos)$; $(iii)$ *for any* $\rho \in \mathcal{F}_{\text{MLP}}$, $\mathcal{N}_{\text{eq.}}(\rho)$ *is dense in* $\mathcal{N}_{\text{eq.}}(\cos)$.

## 4   Numerical illustrations

This section provides numerical illustrations of our findings on simple synthetic examples. The goal is to examine the impact of the tensorization orders $k_s$ and the width $S$. The code is available at https://github.com/nkeriven/univgnn. We emphasize that the contribution of the present paper is first and foremost theoretical, and that, like MLPs with a single hidden layer, we cannot expect the shallow GNNs (1) to be state-of-the-art and compete with deep models, despite their universality. A benchmarking of *deep* GNNs that use invariant and equivariant linear operators is done in (Maron et al., 2019a).

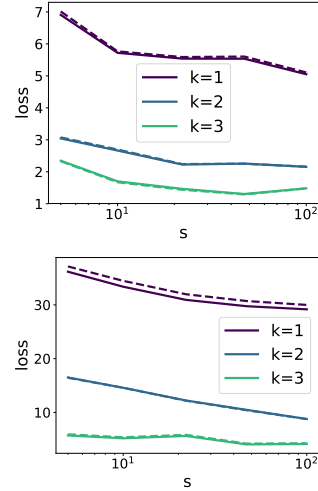

Figure 3: MSE results after 150 epochs, in the invariant (top) and equivariant (bottom) cases, averaged over 5 experiments. Dashed lines represent the testing error.

We consider graphs, represented using their adjacency matrices (i.e. 2-ways tensor, so that $d = 2$). The synthetic graphs are drawn uniformly among 5 graph topologies (complete graph, star, cycle, path or wheel) with edge weights drawn independently as the absolute value of a centered Gaussian variable. Since our approximation results are valid for several graph sizes simultaneously, both training and testing datasets contain $1.4 \cdot 10^4$ graphs, half with 5 nodes and half with 10 nodes. The training is performed by minimizing a square Euclidean loss (MSE) on the training dataset. The minimization is performed by stochastic gradient descent using the ADAM optimizer (Kingma and Ba, 2014). We consider two different regression tasks: (i) in the invariant case, the scalar to predict is the geodesic diameter of the graph, (ii) in the equivariant case, the vector to predict assigns to each node the length of the longest shortest-path emanating from it. While these functions can be computed using polynomial time all-pairs shortest paths algorithms, they are highly non-local, and are thus challenging to learn using neural network architectures. The GNNs (1) are implemented with a fixed tensorization order $k_s = k \in \{1, 2, 3\}$ and $\rho = \rho_{\text{sig.}}$.

Figure 3 shows that, on these two cases, when increasing the width $S$, the out-of-sample prediction error quickly stagnates (and sometime increasing too much $S$ can slightly degrade performances by making the training harder). In sharp contrast, increasing the tensorization order $k$ has a significant impact and lowers this optimal error value. This support the fact that universality relies on the use of higher tensorization order. This is a promising direction of research to integrate higher order tensors withing deeper architecture to better capture complex functions on graphs.

## 5   Conclusion

In this paper, we proved the universality of a class of one hidden layer equivariant networks. Handling this vector-valued setting required to extend the classical Stone-Weierstrass theorem. It remains an open problem to extend this technique of proof for more general equivariant networks whose outputs are graph-valued, which are useful for instance to model dynamic graphs using recurrent architectures (Battaglia et al., 2016). Another outstanding open question, formulated in (Maron et al., 2019b), is the characterization of the approximation power of networks whose tensorization orders $k_s$ inside the layers are bounded, since they are much more likely to be implemented on large graphs in practice.

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
