[Supplementary Material]

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

## Footnotes

[1] If there are $i, j$ such that $x_i < x_j$ and $y_j < y_i$, then we can form $y'$ by swapping $y_i$ and $y_j$, and we have $x^\top y' - x^\top y = x_i y_j + x_j y_i - x_i y_i - x_j y_j = (x_j - x_i)(y_i - y_j) > 0$, hence the swapping strictly increases the scalar product, which is maximal when $x$ and $y$ are ordered in the same fashion.

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

# A   Proofs

## A.1   Invariant case

### A.1.1   Proof of Lemma 1

We first prove that invariant GNNs are continuous with respect to $d_{\text{edit}}$. For two graphs $G_1, G_2$ such that $d_{\text{edit}}(\mathcal{O}(G_1), \mathcal{O}(G_2)) < c$, the graphs have the same number of nodes. Using the fact that $\rho, H, F$ are (locally) Lipschitz in this case, we have $|f(G_1) - f(G_2)| \lesssim \|G_1 - G_2\|_1$, and by invariance by permutation:

$$|f(\mathcal{O}(G_1)) - f(\mathcal{O}(G_2))| \lesssim \min_{\sigma} \|G_1 - \sigma \star G_2\|_1 = d_{\text{edit}}(\mathcal{O}(G_1), \mathcal{O}(G_2))$$

and therefore we have indeed $\mathcal{N}_{\text{inv.}}^{\otimes}(\rho) \subset \mathcal{C}(\mathcal{G}_{\text{inv.}}, d_{\text{edit}})$.

Since $\mathcal{N}_{\text{inv.}}^{\otimes}(\rho)$ is obviously a vector space, we must now prove that it is closed by multiplication. For that, it is sufficient to prove that, for two invariant linear operators $H_1 : \mathbb{R}^{n^{k_1}} \to \mathbb{R}$ and $H_2 : \mathbb{R}^{n^{k_1}} \to \mathbb{R}$, there exists an invariant linear operator $H_3 : \mathbb{R}^{n^{k_1+k_2}} \to \mathbb{R}$ such that $H_1[G_1]H_2[G_2] = H_3[G_1 \otimes G_2]$. For this we recall that $(A \otimes B)(C \otimes D) = (AC) \otimes (BD)$ and $\text{vec}(A) \otimes \text{vec}(B) = \text{vec}(A \otimes B)$, and thus that

$$
\begin{aligned}
H_1[G_1]H_2[G_2] &= \left(\text{vec}(H_1)^\top \text{vec}(G_1)\right)\left(\text{vec}(H_2)^\top \text{vec}(G_2)\right) \\
&= (\text{vec}(H_1)^\top \otimes \text{vec}(H_2)^\top)(\text{vec}(G_1) \otimes \text{vec}(G_2)) \\
&= \text{vec}(H_1 \otimes H_2)^\top \text{vec}(G_1 \otimes G_2)
\end{aligned}
$$

Hence we can define $H_3 = H_1 \otimes H_2$ and check that it is invariant by permutation. By Maron et al. (2019a) a necessary and sufficient condition is $P^{\otimes k_1 + k_2} \text{vec}(H_3) = \text{vec}(H_3)$, which we can easily check:

$$P^{\otimes(k_1+k_2)}\text{vec}(H_3) = (P^{\otimes k_1}\text{vec}(H_1)) \otimes (P^{\otimes k_2}\text{vec}(H_2)) = \text{vec}(H_3)$$

since $P^{\otimes k_i}\text{vec}(H_i) = \text{vec}(H_i)$. $\qquad\square$

### A.1.2   Proof of Lemma 2

We proceed by contradiction, and show that if $f(\mathcal{O}(G)) = f(\mathcal{O}(G'))$ for any $f \in \mathcal{N}_{\text{inv.}}^{\otimes}(\rho_{\text{sig}})$, then $\mathcal{O}(G) = \mathcal{O}(G')$, i.e. $G$ and $G'$ are permutation of each other. Let $G, G'$ be any two such graphs.

The first step if to show that $G$ and $G'$ have the same number of nodes $n = n'$. Consider $\tau = \min_{i_1,\dots,i_d}(\min(G_{i_1,\dots,i_d}, G'_{i_1,\dots,i_d})) - 1$ the minimal element of both $G$ and $G'$ minus 1, and the following family of networks:

$$f_\lambda(G) = H\left[\rho_{\text{sig}}\left(\lambda(G - \tau \mathbf{1}^{\otimes d})\right)\right] \quad \text{with} \quad H[z] = \sum_{i_1,\dots,i_d} z_{i_1,\dots,i_d}.$$

By letting $\lambda \to \infty$, the sigmoid produces 1 for every element in $G$ that is above $\tau$, that is, every element in $G$ or $G'$. Hence we have $f_\lambda(G) \xrightarrow[\lambda\to\infty]{} n^d$ and $f_\lambda(G') \xrightarrow[\lambda\to\infty]{} (n')^d$, and therefore $n = n'$.

Then, we show similarly that the multiset (that is, set with multiplicity) of $\{G_{i_1,\dots,i_d}\}$ is the same as the multiset of $\{G'_{i_1,\dots,i_d}\}$. Consider them ordered: $G_{i_1^{(1)}\dots i_d^{(1)}} \leqslant \dots \leqslant G_{i_1^{(N)}\dots i_d^{(N)}}$, and $G'_{j_1^{(1)}\dots j_d^{(1)}} \leqslant \dots \leqslant G'_{i_1^{(N)}\dots i_d^{(N)}}$, where $N = n^d$. Then, by contradiction, if there is a $q$ such that $G_{i_1^{(q)}\dots i_d^{(q)}} \neq G'_{j_1^{(q)}\dots j_d^{(q)}}$, say $G_{i_1^{(q)}\dots i_d^{(q)}} < G'_{j_1^{(q)}\dots j_d^{(q)}}$ w.l.o.g., set $\tau = (G_{i_1^{(q)}\dots i_d^{(q)}} + G'_{j_1^{(q)}\dots j_d^{(q)}})/2$. Then, for $\lambda > 0$, consider the same neural networks as above with this $\tau$. Again, by letting $\lambda \to \infty$, the sigmoid produces 1 for every element in $G$ that is above $\tau$, and 0 otherwise. Hence $f_\lambda(G) \xrightarrow[\lambda\to\infty]{} n^d - q$, and $f_\lambda(G') \xrightarrow[\lambda\to\infty]{} n^d - q + 1$, which is a contradiction. Hence $G_{i_1^{(q)}\dots i_d^{(q)}} = G'_{j_1^{(q)}\dots j_d^{(q)}}$ for every $q$, and $G$ and $G'$ are formed by the same multiset of $n^d$ real numbers.

Consider now the tensors $A = \rho_{\text{sig}}(G)$, $A' = \rho_{\text{sig}}(G')$ which have strictly positive elements. Since $\rho_{\text{sig}}$ is a 1-to-1 mapping in $\mathbb{R}$, producing a permutation between $A, A'$ yields a permutation for $G, G'$

and allow us to conclude. We consider the following class of neural nets in $\mathcal{N}_{\text{inv.}}^{\otimes}(\rho_{\text{sig}})$:

$$f(G) = H[A^{\otimes k}]$$

for every integer $k > 0$ and invariant $H$. Recall that $A^{\otimes k}$ is an $dk$-order tensor indexed such that

$$(A^{\otimes k})_{(i_{11},\ldots,i_{1d}),\ldots,(i_{k1},\ldots,i_{kd})} = \prod_{\ell=1}^{k} a_{i_{\ell 1},\ldots,i_{\ell d}}$$

for any $1 \leqslant i_{\ell q} \leqslant n$. Then, for any fixed set of such indices, it is not difficult to see that a valid invariant operator is the following:

$$H[A^{\otimes k}] = \sum_{\sigma \in \mathcal{O}_n} \prod_{\ell=1}^{k} a_{\sigma(i_{\ell 1}),\ldots,\sigma(i_{\ell d})}$$

where $\mathcal{O}_n$ is the set of all permutations. Indeed, for all $\bar{\sigma} \in \mathcal{O}_n$:

$$H[(\bar{\sigma} \star A)^{\otimes k}] = \sum_{\sigma \in \mathcal{O}_n} \prod_{\ell=1}^{k} a_{\bar{\sigma}^{-1}\sigma(i_{\ell 1}),\ldots,\bar{\sigma}^{-1}\sigma(i_{\ell d})}$$

$$= \sum_{\sigma \in \mathcal{O}_n} \prod_{\ell=1}^{k} a_{\sigma(i_{\ell 1}),\ldots,\sigma(i_{\ell d})} = H[A^{\otimes k}]$$

by a simple change of variable in the sum $\sum_{\sigma \in \mathcal{O}_n}$. In the same spirit, for any set of integers $k_{i_1,\ldots,i_d} \geqslant 0$ where $1 \leqslant i_q \leqslant n$, the following is a valid invariant GNN in $\mathcal{N}_{\text{inv.}}^{\otimes}(\rho_{\text{sig}})$:

$$f(G) = H[A^{\otimes \sum_{i_1,\ldots,i_d} k_{i_1,\ldots,i_d}}] = \sum_{\sigma \in \mathcal{O}_n} \prod_{i_1,\ldots,i_d=1}^{n} a_{\sigma(i_1),\ldots,\sigma(i_d)}^{k_{i_1,\ldots,i_d}}$$

Hence, we have that for any $k_{i_1,\ldots,i_d}$:

$$\sum_{\sigma \in \mathcal{O}_n} \prod_{i_1,\ldots,i_d=1}^{n} a_{\sigma(i_1),\ldots,\sigma(i_d)}^{k_{i_1,\ldots,i_d}} = \sum_{\sigma \in \mathcal{O}_n} \prod_{i_1,\ldots,i_d=1}^{n} (a')_{\sigma(i_1),\ldots,\sigma(i_d)}^{k_{i_1,\ldots,i_d}}$$

Recalling that $\{a_{i_1,\ldots,i_d}\}$ and $\{a'_{i_1,\ldots,i_d}\}$ are the same multiset, we can apply Lemma 11 in Appendix C, which yields a permutation $\sigma$ such that $a_{i_1,\ldots,i_d} = a'_{\sigma(i_1),\ldots,\sigma(i_d)}$ and concludes the proof. $\qquad \square$

### A.1.3 Proof of Lemma 3

(i) Consider any function in $\mathcal{N}_{\text{inv.}}^{\otimes}(\rho_{\text{sig}})$

$$f(G) = \sum_{s=1}^{S} H_s \left[ \rho_{\text{sig}}(F_{s1}[G] + B_{s1}) \otimes \ldots \otimes \rho_{\text{sig}}(F_{sT_s}[G] + B_{sT_s}) \right] + b$$

and any $\varepsilon > 0$.

Given that we are on a bounded domain, there exists $M$ such that $\sup_G \max_{s,t} \|F_{st}[G] + B_{st}\|_{\infty} \leqslant M$ for all $s$ (where $\|\cdot\|_{\infty}$ is element-wise maximum). The Fourier development of $\rho_{\text{sig}}$ on $[-M, M]$ yields that there exist $a_i, b_i, c_i, i \leqslant N$, such that for all $u \in [-M, M]$

$$\left| \rho_{\text{sig}}(u) - \sum_{i=1}^{N} a_i \cos(b_i u + c_i) \right| \leqslant \varepsilon$$

Defining

$$f_{st}(G) = \rho_{\text{sig}}(F_{st}[G] + B_{st}),$$

$$h_{st}(G) = \sum_{i=1}^{N} a_i \cos\left(b_i(F_{st}[G] + B_{st}) + c_i 1^{\otimes 2k_{st}}\right),$$

we have
$$\sup_G \max_{s,t} \|f_{st}(G) - h_{st}(G)\|_\infty \leqslant \varepsilon$$
Hence, for any $s$, is we define $e_t = \|f_{s1}(G) \otimes \ldots \otimes f_{st}(G) - h_{s1}(G) \otimes \ldots \otimes h_{st}(G)\|_\infty$, we have

$$
\begin{aligned}
e_{T_s} &\leqslant \|f_{s1}(G) \otimes \ldots \otimes f_{sT_s-1}(G) \otimes (f_{sT_s}(G) - h_{sT_s}(G))\|_\infty \\
&\quad + \|((f_{s1}(G) \otimes \ldots \otimes f_{sT_s-1}(G) - h_{s1}(G) \otimes \ldots \otimes h_{sT_s-1}(G)) \otimes h_{sT_s}(G)\|_\infty \\
&\leqslant \varepsilon + (1+\varepsilon)e_{T_s-1} \leqslant 3^{T_s}e_1 \leqslant 3^{T_s}\varepsilon
\end{aligned}
$$

Since the $H_s$ are linear in finite dimension they are bounded operators and we call $L_s$ such that $|H_s(W)| \leqslant L_s \|W\|_\infty$. Finally, if we define $g \in \mathcal{N}^\otimes_{\text{inv.}}(\cos)$ by

$$g(G) = \sum_{s=1}^{S} H_s \left[ h_{s1}(G) \otimes \ldots \otimes h_{sT_s}(G) \right]$$

we have proved that we have $\sup_G |f(G) - g(G)| \leqslant (\sum_s L_s 3^{T_s})\varepsilon$, which concludes the proof.

(ii) The proof is based on the fact that $\cos(a)\cos(b) = \cos(a+b) + \cos(a-b)$. Hence:

$$
\begin{aligned}
\cos(&F_1[G] + B_1) \otimes \cos(F_2[G] + B_2) \\
&= \left( \cos(F_1[G] + B_1) \otimes 1_{n^{k_2}} 1_{n^{k_2}}^\top \right) \odot \left( 1_{n^{k_1}} 1_{n^{k_1}}^\top \otimes \cos(F_2[G] + B_2) \right) \\
&= \cos(\bar{F}_1[G] + \bar{B}_1 + \bar{F}_2[G] + \bar{B}_2) \\
&\quad + \cos(\bar{F}_1[G] + \bar{B}_1 - \bar{F}_2[G] - \bar{B}_2)
\end{aligned}
$$

where $\bar{F}_1[G] = F_1[G] \otimes 1_{n^{k_2}} 1_{n^{k_2}}^\top$ and $\bar{F}_2[G] = 1_{n^{k_1}} 1_{n^{k_1}}^\top \otimes F_2[G]$ and similarly for $\bar{B}_i$. Since $11^\top$ is invariant by permutation, it is easy to see that the $\bar{F}_i$ are equivariant linear functions outputting a $k_1 + k_2$-tensor, and $\bar{B}_i$ are equivariant biases, which proves the result.

(iii) Since $\rho \in \mathcal{F}_{\text{MLP}}$ and the universal approximation theorem applies, the cosine function on a compact of $\mathbb{R}$ can be uniformly approximated by a linear combination of $\rho$:

$$\sup_{x \in [-MM]} \left| \cos(x) - \sum_{i=1}^{N} a_i \rho(b_i x + c_i) \right| \leqslant \varepsilon$$

The rest of the proof is similar to $(i)$.

$\square$

## A.2 Equivariant case

### A.2.1 Proof of Lemma 4

Again we must prove that $\mathcal{N}^\otimes_{\text{eq.}}(\rho)$ is closed by "multiplication", that is, Hadamard product. For that, it is sufficient to show that for two equivariant linear operators $H_1 : \mathbb{R}^{n^k} \to \mathbb{R}^n$, $H_2 : \mathbb{R}^{n^\ell} \to \mathbb{R}^n$, there exists an equivariant linear operator $H_3 : \mathbb{R}^{n^{k+\ell}} \to \mathbb{R}^n$ such that

$$H_1[G_1] \odot H_2[G_2] = H_3[G_1 \otimes G_2]$$

For that, writing the matrices $H_1 \in \mathbb{R}^{n^k \times n}$ and $H_2 \in \mathbb{R}^{n^\ell \times n}$ by abuse of notation, we have

$$H_1[G_1] \odot H_2[G_2] = \text{diag}\left( H_1 \text{vec}\,(G_1)\,\text{vec}\,(G_2)^\top H_2^\top \right)$$

Then, defining $\text{mat}_{k,\ell}$ the operator that transforms a tensor $G \in \mathbb{R}^{n^{k+\ell}}$ to a $\mathbb{R}^{n^k \times n^\ell}$ matrix and the linear operator $H_3[G] = \text{diag}\left( H_1 \text{mat}_{k,\ell}(G) H_2^\top \right)$, we have indeed that $H_1[G_1] \odot H_2[G_2] = H_3[G_1 \otimes G_2]$. Then, for any permutation $\sigma$ and corresponding matrix $P$, since $H_1 P^{\otimes k} = PH_1$, $H_2 P^{\otimes \ell} = PH_2$, and $\text{mat}_{k,\ell}(\sigma \star G) = P^{\otimes k}\text{mat}_{k,\ell}(G)(P^\top)^{\otimes \ell}$, we have

$$
\begin{aligned}
H_3[\sigma \star G] &= \text{diag}\left( H_1 \text{mat}_{k,\ell}(\sigma \star G) H_2^\top \right) \\
&= \text{diag}\left( H_1 P^{\otimes k} \text{mat}_{k,\ell}(G)(P^\top)^{\otimes \ell} H_2^\top \right) \\
&= \text{diag}\left( PH_1 \text{mat}_{k,\ell}(G) H_2^\top P^\top \right) = PH_3[G]
\end{aligned}
$$

and therefore $H_3$ is equivariant, which concludes the proof. $\square$

### A.2.2 Proof of Lemma 5

**Separability.** The separability condition is in fact exactly equivalent to the invariant case: indeed, we can construct linear equivariant operators $H_s$ just by stacking linear invariant operators on every coordinate. Hence, for any invariant GNN $f \in \mathcal{N}_{\text{inv.}}^{\otimes}(\rho_{\text{sig.}})$, $h = [f, \dots, f] \in \mathcal{N}_{\text{eq.}}^{\otimes}(\rho_{\text{sig.}})$ is a valid equivariant operator. Hence, for any two graphs $G, G'$ such that are not permutation of each other, by Lemma 2 there is $f \in \mathcal{N}_{\text{inv.}}^{\otimes}(\rho_{\text{sig.}})$ such that $f(G) \neq f(G')$, and by considering $h = [f, \dots, f]$ every coordinate of $h(G)$ is different from that of $h(G')$.

**Self-separability.** For the self-separability, consider any $G \in \mathcal{G}_{\text{eq.}}$ with $n$ nodes, and any $I \subset [n]$. Once again we proceed by contradiction: we are going to show that if there exist $k \in I, \ell \in I^c$ such that for all $h \in \mathcal{N}_{\text{eq.}}^{\otimes}(\rho_{\text{sig.}})$ we have $[h(G)]_k = [h(G)]_\ell$, then $G \in \mathcal{G}_{\text{eq.}}(\mathcal{O}_I)$. Let $G$ be such a graph, with the corresponding fixed $k, \ell$.

Similar to the proof of the separability in the invariant case, we define $A = \rho_{\text{sig.}}(G)$, again keeping in mind that the sigmoid in a one-to-one mapping. Then, for any $k_{i_1, \dots, i_d}$, recall that the following is a valid *invariant* GNN:

$$H[A^{\otimes \sum_{i_1, \dots, i_d} k_{i_1, \dots, i_d}}] = \sum_{\sigma \in \mathcal{O}_n} \prod_{i_1, \dots, i_d = 1}^{n} a_{\sigma(i_1), \dots, \sigma(i_d)}^{k_{i_1, \dots, i_d}}$$

Similarly, we are going to show that the following defines a valid equivariant GNN:

$$[f(G)]_q = \left[ H[A^{\otimes \sum_{i_1, \dots, i_d} k_{i_1, \dots, i_d}}] \right]_q = \sum_{\sigma \in \mathcal{O}^{(q)}} \prod_{i_1, \dots, i_d = 1}^{n} a_{\sigma(i_1), \dots, \sigma(i_d)}^{k_{i_1, \dots, i_d}}$$

where $\mathcal{O}^{(q)} \overset{\text{def.}}{=} \{\sigma \in \mathcal{O} \; ; \; \sigma(k) = q\}$ (where we recall that $k, \ell$ are fixed and part of the hypothesis we have made on $G$). Indeed, for any permutation $\bar{\sigma}$, we have

$$[f(\bar{\sigma} \star G)]_{\bar{\sigma}(q)} = \sum_{\sigma \in \mathcal{O}^{(\sigma(q))}} \prod_{i_1, \dots, i_d = 1}^{n} a_{\bar{\sigma}^{-1}(\sigma(i_1)), \dots, \bar{\sigma}^{-1}(\sigma(i_d))}^{k_{i_1, \dots, i_d}}$$

$$= \sum_{\sigma \in \mathcal{O}, \bar{\sigma}^{-1}(\sigma(k)) = q} \prod_{i_1, \dots, i_d = 1}^{n} a_{\bar{\sigma}^{-1}(\sigma(i_1)), \dots, \bar{\sigma}^{-1}(\sigma(i_d))}^{k_{i_1, \dots, i_d}}$$

$$= \sum_{\sigma \in \mathcal{O}, \sigma(k) = q} \prod_{i_1, \dots, i_d = 1}^{n} a_{\sigma(i_1), \dots, \sigma(i_d)}^{k_{i_1, \dots, i_d}} = [f(G)]_q$$

Hence, we have indeed $f(\bar{\sigma} \star G) = \bar{\sigma} \star f(G)$, and $f$ is equivariant. Now, by hypothesis on $G$, it means that for all $k_{i_1, \dots, i_d}$, we have:

$$\sum_{\sigma \in \mathcal{O}^{(k)}} \prod_{i_1, \dots, i_d = 1}^{n} a_{\sigma(i_1), \dots, \sigma(i_d)}^{k_{i_1, \dots, i_d}} = \sum_{\sigma \in \mathcal{O}^{(\ell)}} \prod_{i_1, \dots, i_d = 1}^{n} a_{\sigma(i_1), \dots, \sigma(i_d)}^{k_{i_1, \dots, i_d}} .$$

Now, since $\mathcal{O}^{(k)}$ contains the identity and has the same cardinality as $\mathcal{O}^{(\ell)}$, by Lemma 11 it means that there is a permutation $\sigma \in \mathcal{O}^{(\ell)}$ such that $G = \sigma \star G$. Observing that $\mathcal{O}^{(\ell)} \subset \mathcal{O}_I$ concludes the proof. $\qquad \square$

## B Adapted Stone-Weierstrass theorem: proof of Theorem 4

Let us first introduce some more notations. For $\mathcal{O}' \subset \mathcal{O}$ and $\mathcal{G}$ a set of graphs, we define

$$\mathcal{O}'(\mathcal{G}) \overset{\text{def.}}{=} \{\sigma \star G \; ; \; \sigma \in \mathcal{O}', G \in \mathcal{G}\}$$

$$\mathcal{G}(\mathcal{O}') \overset{\text{def.}}{=} \{G \in \mathcal{G} \; ; \; \exists \sigma \in \mathcal{O}', G = \sigma \star G\}$$

that is, respectively, the set of permuted graphs in $\mathcal{G}$, and the set of graphs in $\mathcal{G}$ that have a self-isomorphism in $\mathcal{O}'$. Recall that we denote by $[f]_I$ and $[x]_I$ indexation of multivariate functions and vectors, and that inequalities $x \geqslant a$ are element-wise. A neighborhood of $x$ is an open set $V$ such that $x \in V$. Finally, for convenience we denote $\mathcal{G}_{\text{eq.}}^{(n)} = \mathcal{G}_{\text{eq.}} \cap \mathbb{R}^{n^d}$ the graphs in $\mathcal{G}_{\text{eq.}}$ that have $n$ nodes.

As described in the paper, the key lemma is the construction of *step-functions*.

**Lemma 7** (Existence of step-functions). *Let $n \leqslant n_{\max}$, and $I \subset [n]$ be any subset of indices. Let $A \subset \mathcal{G}_{\text{eq.}}, B \subset \mathcal{G}_{\text{eq.}}^{(n)}$ be two closed sets such that $B \cap \mathcal{G}_{\text{eq.}}^{(n)}(\mathcal{O}_I) = \emptyset$ and $B \cap \mathcal{O}(A) = \emptyset$, that is, graphs in $B$ have no self-isomorphism in $\mathcal{O}_I$ and no two graphs between $A$ and $B$ are isomorphic. Then, for all $\varepsilon > 0$, there exists $f \in \mathcal{A}$ such that:*

$$\begin{cases} \forall G, & 0 \leqslant f(G) \leqslant 1 \\ \forall G \in B, & [f(G)]_I \geqslant 1 - \varepsilon \quad and \quad [f(G)]_{I^c} \leqslant \varepsilon \\ \forall G \in A, & f(G) \leqslant \varepsilon \end{cases}$$

We start the proof by a serie of three intermediate lemmas.

**Lemma 8.** *Let $n \leqslant n_{\max}$, and $I \subset [n]$ be any subset of indices. Let $G_0 \in \mathcal{G}_{\text{eq.}}^{(n)}$ such that $G \notin \mathcal{G}_{\text{eq.}}^{(n)}(\mathcal{O}_I)$, and $T$ be a closed subset of $\mathcal{G}_{\text{eq.}}$ such that $T \cap \mathcal{O}(G_0) = \emptyset$. Then, there exists $V(G_0) \subset \mathbb{R}^{n^d}$ a neighborhood of $G_0$ such that the following holds: for all $\varepsilon > 0$, there exists $f \in \mathcal{A}$ such that:*

$$\begin{cases} \forall G, & f(G) \in [0,1] \\ \forall G \in V(G_0), & [f(G)]_I \geqslant 1 - \varepsilon \quad and \quad [f(G)]_{I^c} \leqslant \varepsilon \\ \forall G \in T, & f(G) \leqslant \varepsilon \end{cases}$$

*Proof.* Our goal is to build a function $g \in \mathcal{A}$ along with a threshold $\delta > 0$ and $V(G_0)$ a neighborhood of $G_0$ such that:

$$\begin{cases} \forall G \in T, & g(G) \geqslant \delta \\ \forall G \in V(G_0), & [g(G)]_I \leqslant \delta/2 \text{ and } [g(G)]_{I^c} \geqslant \delta \end{cases}$$

Then we can conclude similarly to the end of the proof of Lemma 1 in (Brosowski and Deutsch, 1981).

Take any $k \in I, \ell \in I^c$ and $G \in T$. Note that $G$ does not necessarily have $n$ nodes, we denote $n_G$ its number of nodes. Let $i \in [n_G]$ be any index. According to the two separability hypotheses, there exists $g_{G,k,i}, h_{k,\ell} \in \mathcal{A}$ such that $[g_{G,k,i}(G_0)]_k \neq [g_{G,k,i}(G)]_i$ and $[h_{k,\ell}(G_0)]_k \neq [h_{k,\ell}(G_0)]_\ell$. Then, consider

$$g_G = \prod_{k \in I} \left( \frac{1}{n_G} \sum_{i=1}^{n_G} \frac{(g_{G,k,i} - [g_{G,k,i}(G_0)]_k \mathbf{1})^2}{\|g_{G,k,i} - [g_{G,k,i}(G_0)]_k \mathbf{1}\|_\infty^2} \right) \in \mathcal{A}$$

$$h = \prod_{k \in I} \left( \frac{1}{|I^c|} \sum_{\ell \in I^c} \frac{(h_{k,\ell} - [h_{k,\ell}(G_0)]_k \mathbf{1})^2}{\|h_{k,\ell} - [h_{k,\ell}(G_0)]_k \mathbf{1}\|_\infty^2} \right) \in \mathcal{A}$$

where $\prod, (\cdot)^2$ are to be understood component-wise and $\|g\|_\infty = \sup_G \|g(G)\|_\infty$. These functions satisfy

$$\begin{cases} g_G, h \in [0,1], \\ [g_G(G_0)]_I = [h(G_0)]_I = 0 \\ g_G(G) > 0, [h(G_0)]_{I^c} > 0 \end{cases}$$

By continuity, define $S(G) \subset \mathbb{R}^{n_G^d}$ a neighborhood of $G$ such that $g_G > 0$ on $S(G)$. By compacity of $T$, there is a finite number of $G_1, \ldots, G_m$ such that $T \subset \bigcup_i S(G_i)$. Then, we define $g = \frac{1}{m+1}(\sum_i g_{G_i} + h) \in \mathcal{A}$, which satisfies:

$$\begin{cases} g \in [0,1] \\ g > 0 \text{ on } T \\ [g(G_0)]_I = 0 \text{ and } [g(G_0)]_{I^c} > 0 \,. \end{cases}$$

Again, by compacity of $T$, there exists $\delta > 0$ such that $g \geqslant \delta$ on $T$ and $[g(G_0)]_{I^c} \geqslant 2\delta$. Then, by continuity, we define $V(G_0)$ a neighborhood of $G_0$ such that $[g]_I \leqslant \delta/2$ and $[g]_{I^c} \geqslant \delta$ on $V(G_0)$.

We can now conclude. Assuming that $\delta < 1$ is small enough without lost of generality, let $k$ be an integer such that $1/\delta < k < 2/\delta$, and define the following functions in $\mathcal{A}$:

$$q_p = (1 - g^p)^{k^p}$$

which are obviously such that $q_p \in [0, 1]$.

Then, using the elementary Bernoulli inequality $(1 + h)^p \geqslant 1 + ph$ for all $h \geqslant -1$, we have for all $G \in V(G_0)$ and $i \in I$:

$$q_p(G)_i \geqslant 1 - (kg_i(G))^p \geqslant 1 - (k\delta/2)^p \xrightarrow[p\to\infty]{} 1$$

and similarly, for either $G \in T$ and any $i$, or $G \in V(G_0)$ and $i \in I^c$, we have

$$q_p(G)_i \leqslant \frac{1 + (kg_i(G))^p}{(kg_i(G))^p}(1 - g_i(G)^p)^{k^p} \leqslant \frac{(1 + g_i(G)^p)^{k^p}}{(kg_i(G))^p}(1 - g_i(G)^p)^{k^p} \text{ by Bernoulli's inequality}$$

$$= \frac{(1 - g_i(G)^{2p})^{k^p}}{(kg_i(G))^p} \leqslant \frac{1}{(k\delta)^p} \xrightarrow[p\to\infty]{} 0$$

Hence, for all $\varepsilon > 0$, there exists an $p$ such that $q_p \leqslant \varepsilon$ on $T$, $[q_p]_{I^c} \leqslant \varepsilon$ and $[q_p]_I \geqslant 1 - \varepsilon$ on $V(G_0)$. Taking $f = 1 - q_p$ concludes the proof. □

A similar result without the interval $I$ is the following.

**Lemma 9.** *Let $G_0$ be any graph and $T$ be a closed subset of $\mathcal{G}_{eq.}$ such that $T \cap \mathcal{O}(G_0) = \emptyset$. Then, there exists $V(G_0)$ a neighborhood of $G_0$ such that the following holds: for all $\varepsilon > 0$, there exists $f \in \mathcal{A}$ such that:*

$$\begin{cases} \forall G, & f(G) \in [0, 1] \\ \forall G \in V(G_0), & f(G) \geqslant 1 - \varepsilon \\ \forall G \in T, & f(G) \leqslant \varepsilon \end{cases}$$

*Proof.* The proof is similar (but simpler) to that of Lemma 8, without introduction of the interval $I$ and the function $h$. □

An easy consequence of the above Lemma is the following.

**Lemma 10.** *Let $A, B$ be two closed sets such that $B \cap \mathcal{O}(A) = \emptyset$. Then, for all $\varepsilon > 0$, there exists $f \in \mathcal{A}$ such that:*

$$\begin{cases} \forall G, & f(G) \in [0, 1] \\ \forall G \in B, & f(G) \geqslant 1 - \varepsilon \\ \forall G \in A, & f(G) \leqslant \varepsilon \end{cases}$$

*Proof.* Let $G \in B$. By hypothesis, $A \cap \mathcal{O}(G) = \emptyset$, so by Lemma 9 there exists $V(G)$ a neighborhood of $G$ such that for all $\varepsilon > 0$ there exists $f \in \mathcal{A}$ satisfying: $0 \leqslant f \leqslant 1$, $f \geqslant 1 - \varepsilon$ on $V(G)$, and $f \leqslant \varepsilon$ on $A$. By compacity of $B$, there is a finite number of $G_1, \ldots, G_m \in B$ such that $B \subset \cup_{i=1}^m V(G_i)$. Denote by $f_i$ the associated functions produced by Lemma 9 for some $\varepsilon' > 0$, and denote $f = \prod_i (1 - f_i)$. We have that $f \leqslant \varepsilon'$ on $B$ and $f \geqslant (1 - \varepsilon')^m$ on $A$. Hence by choosing appropriately $\varepsilon'$ (note that $\varepsilon'$ is authorized to depend on $m$), we obtain a function $f$ such that $f \leqslant \varepsilon$ on $B$ and $f \geqslant 1 - \varepsilon$ on $A$, and taking $1 - f$ concludes the proof. □

We can now show Lemma 7.

*Proof of Lemma 7.* Let $G \in B \subset \mathcal{G}_{eq.}^{(n)}$. By hypothesis, $G \notin \mathcal{G}_{eq.}^{((n))}(\mathcal{O}_I)$ and $A \cap \mathcal{O}(G) = \emptyset$, so by Lemma 8 there exists $V(G) \subset \mathbb{R}^{n^d}$ a neighborhood of $G$ such that for all $\varepsilon > 0$ there exists $f \in \mathcal{A}$ satisfying:

$$0 \leqslant f \leqslant 1$$
$$[f]_I \geqslant 1 - \varepsilon \text{ and } [f]_{I^c} \leqslant \varepsilon \text{ on } V(G)$$
$$f \leqslant \varepsilon \text{ on } A.$$

By compacity of $B$, there is a finite number of $G_1, \ldots, G_m \in B$ such that $B \subset \cup_{i=1}^m V(G_i)$. For some $\varepsilon > 0$ that we will choose later, denote the associated functions $f_1, \ldots, f_m$ (note that the $V(G_i)$ do not depend on $\varepsilon$, but the $f_i$ do).

We remark that we cannot just consider the function $\prod_i f_i$ and conclude: indeed, on each $V(G_i)$ only $f_i$ will satisfy $[f_i]_I \geqslant 1 - \varepsilon$, and the others $f_j$ are not guaranteed to be lower bounded. For the same reason, we cannot consider $\frac{1}{m}\sum_i f_i$ either, due to the requirement that $[f]_{I^c} \leqslant \varepsilon$ on $B$. We need to introduce auxiliary functions $\tilde{f}_i$ such that we are guaranteed that for each $j \neq i$, $[\tilde{f}_j]_I \geqslant 1 - \varepsilon$ on $V(G_i)$, and we can conclude with $\prod_i (f_i + \tilde{f}_i)$. We will construct such functions with Lemma 10. A final difficulty is that $V(G_i)$ are open sets, while Lemma 10 can only work with closed sets.

Hence, by continuity, consider the neighborhoods $V'(G_i) \subset \mathbb{R}^{n^d}$ such that

$$\overline{V(G_i)} \subset V'(G_i)$$
$$[f_i]_I \geqslant 1 - 2\varepsilon \text{ and } [f_i]_{I^c} \leqslant 2\varepsilon \text{ on } V'(G_i).$$

Note that the $V(G_i)$ do not depend on $\varepsilon$, but the $V'(G_i)$ do.

Then, for all $i \in [n]$ consider the closed sets $A_i = A \cup \overline{V(G_i)}$ and $B_i = B \backslash \mathcal{O}(V'(G_i))$. By construction of $B_i$ and hypothesis on $A$, we have indeed that $\mathcal{O}(A_i) \cap B_i = \emptyset$, since $\overline{V(G_i)} \subset V'(G_i)$. Applying Lemma 10, we obtain a function $\tilde{f}_i \in \mathcal{A}$ such that $\tilde{f} \leqslant \varepsilon$ on $A_i$ and $\tilde{f} \geqslant 1 - \varepsilon$ on $B_i$.

Finally, consider the following function: $f = \frac{1}{2^m}\prod_i (f_i + \tilde{f}_i)$. Take any $G \in B$. Consider the index $i$ such that $G \in V(G_i)$. We have $[f_i(G) + \tilde{f}_i(G)]_I \geqslant 1 - \varepsilon$ by definition of $f_i$ and $[f_i(G) + \tilde{f}_i(G)]_{I^c} \leqslant 2\varepsilon$ by definition of $f_i$ and $\tilde{f}_i$ and the fact that $G \in V(G_i) \subset A_i$. For any $j \neq i$, we have the following: either $G \in \mathcal{O}(V'(G_j))$, in which case, by equivariance of $f_j$ and the fact that $G \notin \mathcal{G}_{\text{eq.}}(\mathcal{O}_I)$, we have $[f_j(G)]_I \geqslant 1 - 2\varepsilon$; or $G \in B_j$, in which case $[\tilde{f}_j(G)]_I \geqslant 1 - 2\varepsilon$. Overall, we obtain that

$$[f]_I \geqslant \frac{1}{2^m}(1 - 2\varepsilon)^m \text{ and } [f]_{I^c} \leqslant \frac{1}{2^m}2\varepsilon \text{ on } B$$

$$f \leqslant \frac{1}{2^m}2\varepsilon \text{ on } A.$$

We conclude by choosing $\varepsilon$ such that $(1 - 2\varepsilon)^m > 2\varepsilon$ and proceeding similarly to the end of the proof of Lemma 8, resorting to Bernoulli's inequality. $\qquad \square$

We are now ready to prove Theorem 4.

*Proof of Theorem 4.* Fix $f \in \mathcal{C}_{\text{eq.}}$ a continuous equivariant function and $\varepsilon > 0$. Our goal is to find a function $g \in \mathcal{A}$ such that for all $G \in \mathcal{G}_{\text{eq.}}$, $\|F(G) - f(G)\|_\infty \leqslant \varepsilon$. Since $\mathcal{G}_{\text{eq.}}$ is compact, $f$ is bounded, and since we can add constants to $g$, without lost of generality we assume that $0 < f < f_{\max}$ on $\mathcal{G}_{\text{eq.}}$.

We first restrict the space to the compact set where the coordinates of $F$ are ordered:

$$\mathcal{G}_f \stackrel{\text{def.}}{=} \bigcup_{n=1}^{n_{\max}} \mathcal{G}_f^{(n)} \quad \text{where} \quad \mathcal{G}_f^{(n)} \stackrel{\text{def.}}{=} \left\{ G \in \mathcal{G}_{\text{eq.}}^{(n)} \; ; \; f_1(G) \geqslant f_2(G) \geqslant \ldots \geqslant f_n(G) \right\}$$

Indeed, by equivariance of $f$, every graph $G \in \mathcal{G}_{\text{eq.}}$ has a permuted representation in $\mathcal{G}_f$. Hence proving the uniform approximation of $f$ on $\mathcal{G}_f$ is sufficient to prove it on the whole set $\mathcal{G}_{\text{eq.}}$.

Now, denote $K \in \mathbb{N}$ an integer such that $(K - 1)\varepsilon \leqslant f_{\max} \leqslant K\varepsilon$. For $k = 1, \ldots, K$, $n = 1, \ldots, n_{\max}$ and $\ell = 1, \ldots, n$, define the following compact set:

$$A_k^{n,\ell} = \left\{ G \in \mathcal{G}_f^{(n)} \; ; \; f_\ell(G) - f_{\ell+1}(G) \leqslant (k - 1/3)\varepsilon \right\} \cup \bigcup_{n' \neq n} \mathcal{G}_f^{(n')}$$

$$B_k^{n,\ell} = \left\{ G \in \mathcal{G}_f^{(n)} \; ; \; f_\ell(G) - f_{\ell+1}(G) \geqslant (k + 1/3)\varepsilon \right\}$$

where we use the convention that for $G \in \mathbb{R}^{n^d}$, $f_{n+1}(G) = 0$. Note that $A_k^{n,\ell} \subset A_{k+1}^{n,\ell}$, and $B_{k+1}^{n,\ell} \subset B_k^{n,\ell}$. For $\ell = 1, \ldots, n$ we denote the integer interval $I_\ell = [1, \ell]$.

Let us first show that $A_k^{n,\ell} \cap \mathcal{O}(B_k^{n,\ell}) = \emptyset$ and $B_k^\ell \cap \mathcal{G}_{\text{eq.}}^{(n)}(\mathcal{O}_{I_\ell}) = \emptyset$, so that we can apply Lemma 7. Consider $G \in B_k^{n,\ell}$, $G' \in A_k^{n,\ell}$, $y = F(G)$ and $y' = F(G')$. If $G' \in \mathcal{G}_{\text{eq.}}^{(n')}$ for $n' \neq n$, $G$ and

$G'$ are obviously not permutation of one another. If $G' \in \mathcal{G}_{\text{eq.}}^{(n)}$, the coordinates of both $y$ and $y'$ are sorted, and we have $y'_\ell - y'_{\ell+1} > y_\ell - y_{\ell+1}$ (again with the convention that $y_{n+1}, y'_{n+1} = 0$). It is therefore impossible for $y$ and $y'$ to be permutation of one another, and thus $A_k^{n,\ell} \cap \mathcal{O}(B_k^{n,\ell}) = \emptyset$. Furthermore, for any $\ell < n$, we have $y_i \geqslant y_\ell > y_{\ell+1} \geqslant y_j$ for any $i \leqslant \ell < j$, and therefore there is no self-permutation of $y$ that exchange an index before $\ell$ and one after. In other words, we have $B_k^{n,\ell} \cap \mathcal{G}_{\text{eq.}}^{(n)}(\mathcal{O}_{I_\ell}) = \emptyset$.

Then, for all $k$ and $n$, for $\ell < n$, by applying Lemma 7 for $\varepsilon > 0$ we obtain $f_k^{n,\ell}$ such that $0 \leqslant f_k^{n,\ell} \leqslant 1$, $[f_k^{n,\ell}]_{I_\ell} \geqslant 1 - \varepsilon$ and $[f_k^{n,\ell}]_{I_\ell^c} \leqslant \varepsilon$ on $B_k^{n,\ell}$, and $f_k^{n,\ell} \leqslant \varepsilon$ on $A_k^{n,\ell}$. Similarly, by applying Lemma 10, we obtain functions $f_k^{n,n} \in \mathcal{A}$ such that $0 \leqslant f_k^{n,n} \leqslant 1$, $f_k^{n,n} \geqslant 1 - \varepsilon$ on $B_k^{n,n}$ and $f_k^{n,n} \leqslant \varepsilon$ on $A_k^{n,n}$. Finally, we define $g = \sum_{k,n} \sum_{\ell=1}^n f_k^{n,\ell}$.

Now, take any $G \in \mathcal{G}_f$, denote by $n$ its number of nodes. For every $\ell \leqslant n$, denote $k_\ell$ such that $k_\ell - \frac{2}{3} \leqslant \frac{f_\ell(G) - f_{\ell+1}(G)}{\varepsilon} \leqslant k_\ell + \frac{2}{3}$. By summing these equations, we obtain for all $q$:

$$\varepsilon \sum_{\ell=q}^n k_\ell - \frac{2n\varepsilon}{3} \leqslant \varepsilon \sum_{\ell=q}^n \left( k_\ell - \frac{2}{3} \right) \leqslant (f(G))_q \leqslant \varepsilon \sum_{\ell=q}^n \left( k_\ell + \frac{2}{3} \right) \leqslant \varepsilon \sum_{\ell=q}^n k_\ell + \frac{2n\varepsilon}{3} \quad (4)$$

We are going to show that $g$ approximates these bounds. For all $\ell$, we have $G \in A_{k_\ell+1}^{n,\ell} \cup B_{k_\ell-1}^{n,\ell}$. Moreover, it is obvious that $G \in A_k^{n',\ell}$ for all $n' \neq n$, and all $k, \ell$. By construction of $f_k^{n,\ell}$, we have:

$$\forall n' \neq n, \forall k, \ell, f_k^{n',\ell}(G) \leqslant \varepsilon \qquad\qquad \text{since } G \in A_k^{n',\ell}$$

$$\forall \ell \leqslant n, \forall k \geqslant k_\ell + 1, \quad f_k^{n,\ell}(G) \leqslant \varepsilon \qquad\qquad \text{since } G \in A_{k_\ell+1}^{n,\ell} \subset A_k^{n,\ell}$$

$$\forall \ell \leqslant n, \forall k \leqslant k_\ell - 1, \begin{cases} [f_k^{n,\ell}(G)]_{[1,\ell]} & \geqslant 1 - \varepsilon \\ [f_k^{n,\ell}(G)]_{[\ell+1,n]} & \leqslant \varepsilon \end{cases} \qquad \text{since } G \in B_{k_\ell-1}^{n,\ell} \subset B_k^{n,\ell}$$

Then, we decompose

$$[f(G)]_q = \varepsilon \left( \sum_{n' \neq n} \sum_{\ell=1}^{n'} \sum_k [f_k^{n',\ell}(G)]_q + \sum_{\ell=1}^{q-1} \sum_k [f_k^{n,\ell}(G)]_q + \sum_{\ell=q}^n \sum_k [f_k^{n,\ell}(G)]_q \right) \quad (5)$$

By what precedes the first term is bounded by

$$0 \leqslant \sum_{n' \neq n} \sum_{\ell=1}^{n'} \sum_k [f_k^{n',\ell}(G)]_q \leqslant n_{\max}^2 K\varepsilon < n_{\max}^2 f_{\max}$$

For the second term, we have

$$0 \leqslant \sum_{\ell=1}^{q-1} \sum_k [f_k^{n,\ell}(G)]_q \leqslant (q-1)(1 + (K-1)\varepsilon) < n_{\max}(1 + f_{\max})$$

since for all $\ell \leqslant q - 1$ and any $k \neq k_q$, we have $[f_k^{n,\ell}(G)]_q \leqslant \varepsilon$. Finally, for the third term:

$$\sum_{\ell=q}^n \sum_k [f_k^{n,\ell}(G)]_q \geqslant (1-\varepsilon) \sum_{\ell=q}^n (k_\ell - 1) \geqslant \sum_{\ell=q}^n k_\ell - n_{\max}(f_{\max} + 1),$$

$$\sum_{\ell=q}^n \sum_k [f_k^{n,\ell}(G)]_q \leqslant \sum_{\ell=q}^n (k_\ell + 1 + (K - k_\ell - 1)\varepsilon) \leqslant \sum_{\ell=q}^n k_\ell + n_{\max}(1 + f_{\max})$$

Hence, combining (4) and (5) with the bounds above we obtain

$$-\varepsilon \left( \frac{2n_{\max}}{3} + n_{\max}(f_{\max} + 1) \right) \leqslant [f(G)]_q - [F(G)]_q \leqslant \varepsilon \left( \frac{2n_{\max}}{3} + 2n_{\max}(1 + f_{\max}) + n_{\max}^2 f_{\max} \right)$$

Hence appropriately choosing $\varepsilon$ concludes the proof of the theorem. $\qquad\square$

## C   Additional technical lemma

The next technical lemma is used in proving separation of points.

**Lemma 11.** *Let $a_{i_1,\ldots,i_d}, a'_{i_1,\ldots,i_d} > 0$ for $1 \leqslant i_1,\ldots,i_d \leqslant n$ be $n^d$ positive numbers such that $\{a_{i_1,\ldots,i_d}\}$ and $\{a'_{i_1,\ldots,i_d}\}$ are the same multisets. Let $\mathcal{O}', \mathcal{O}'' \subset \mathcal{O}_n$ be sets of permutations such that $\mathrm{Id} \in \mathcal{O}'$ and $|\mathcal{O}'| = |\mathcal{O}''|$. If, for every set of integers $k_{i_1,\ldots,i_d} > 0$, we have*

$$\sum_{\sigma \in \mathcal{O}'} \prod_{i_1,\ldots,i_d} a_{\sigma(i_1),\ldots,\sigma(i_d)}^{k_{i_1,\ldots,i_d}} = \sum_{\sigma \in \mathcal{O}''} \prod_{i_1,\ldots,i_d} (a'_{\sigma(i_1),\ldots,\sigma(i_d)})^{k_{i_1,\ldots,i_d}} , \tag{6}$$

*then there exists a permutation $\sigma \in \mathcal{O}''$ such that $a_{i_1,\ldots,i_d} = a'_{\sigma(i_1),\ldots,\sigma(i_d)}$.*

*Proof.* The proof is ultimately based on the simple fact that for two vectors $x, y \in \mathbb{R}^p$, the inner product $\langle x, y \rangle$ is maximum if the elements of $x$ and $y$ have the same ordering (the largest element $x_i$ is multiplied to the largest element $y_i$, and so on[1]).

Let us begin by fixing any $k_{i_1,\ldots,i_d}$ and showing that there is a bijection $\varphi : \mathcal{O}' \to \mathcal{O}''$ between the two considered sets of permutations such that for all $\sigma \in \mathcal{O}'$, $\prod_{i_1,\ldots,i_d} a_{\sigma(i_1),\ldots,\sigma(i_d)}^{k_{i_1,\ldots,i_d}} = \prod_{i_1,\ldots,i_d} (a'_{\varphi(\sigma)(i_1),\ldots,\varphi(\sigma)(i_d)})^{k_{i_1,\ldots,i_d}}$, that is, there is a bijection between each additive term of (6). Denoting $A_\sigma = \prod_{i_1,\ldots,i_d} a_{\sigma(i_1),\ldots,\sigma(i_d)}^{k_{i_1,\ldots,i_d}}$ and similarly $A'_\sigma$ for $a'$, a consequence of (6) is that

$$\sum_{\sigma \in \mathcal{O}'} A_\sigma^k = \sum_{\sigma \in \mathcal{O}''} (A'_\sigma)^k \tag{7}$$

for all $k$. Hence, considering the maximum elements $\max_\sigma A_\sigma$ and $\max_{\sigma'} A'_{\sigma'}$ (with arbitrary choice in case of multiple maxima): if they are different, by dividing the equation by the largest of the two and letting $k \to \infty$, we have that one side goes to $0$ while the other tends to a positive constant. Hence the maximal elements are the same, we can substract them from the equation and reiterate. Hence we have proven that there is indeed a bijection between the $A_\sigma$ and $A'_{\sigma'}$.

Then, considering the multiset of $n^d$ numbers $\{a_{i_1,\ldots,i_d}\}$, pick $k_{i_1,\ldots,i_d} > 0$ in the same order than these numbers: $a_{i_1,\ldots,i_d} \leqslant a_{i'_1,\ldots,i'_d}$ implies $k_{i_1,\ldots,i_d} \leqslant k_{i'_1,\ldots,i'_d}$. Using the previously proved property, consider the permutation $\sigma \in \mathcal{O}''$ (which depends on the $k_{i_1,\ldots,i_d}$) such that

$$\prod_{i_1,\ldots,i_d} a_{i_1,\ldots,i_d}^{k_{i_1,\ldots,i_d}} = \prod_{i_1,\ldots,i_d} (a'_{\sigma(i_1),\ldots,\sigma(i_d)})^{k_{i_1,\ldots,i_d}}$$

(that is, we have isolated the term corresponding to $\mathrm{Id} \in \mathcal{O}'$ in the l.h.s. of (7) and located the component in the r.h.s. that is in bijection with it). Then, remembering that the $a_{i_1,\ldots,i_d}$ and $a'_{i_1,\ldots,i_d}$ are taken from the same pool of real numbers, we claim that having the $a'_{\sigma(i_1),\ldots,\sigma(i_d)}$ ordered as the $k_{i_1,\ldots,i_d}$ is the only way to reach the maximum value (reached by the $a_{i_1,\ldots,i_d}$) among all orderings of the $\{a'_{\sigma(i_1),\ldots,\sigma(i_d)}\}$: indeed, for that, take the logarithm of the equation above, and we use the fact that the scalar product between two vectors formed by a fixed set of elements is maximal when they are ordered in the same fashion. Hence we have proven that: $a_{i_1,\ldots,i_d} \leqslant a_{i'_1,\ldots,i'_d}$ implies $k_{i_1,\ldots,i_d} \leqslant k_{i'_1,\ldots,i'_d}$ which implies $a'_{\sigma(i_1),\ldots,\sigma(i_d)} \leqslant a'_{\sigma(i'_1),\ldots,\sigma(i'_d)}$. Since the $a, a'$ are drawn from the same pool of numbers, we have proven that $a_{i_1,\ldots,i_d} = a'_{\sigma(i_1),\ldots,\sigma(i_d)}$, which concludes the proof.   $\square$