[Reviews · NeurIPS 2019]

Reviewer 1



This work proves that any continuous equivariant function of graphs can be approximated by a one hidden layer graph neural network that uses higher-order tensorization and an arbitrary squashing function. One of the main contributions of the paper is the introduction of the framework based on the Stone-Weierstrass theorem, which can be used to reprove a version of a previously known equivalent for invariant graph neural networks. On the negative side, this framework seems to lead to fewer insights such as the required order of tensorization and whether the underlying permutation group can be different from the set of all permutations. Personally, I have found the mathematical part interesting and this result can be accepted to NeurIPS but given what we have already known about invariant functions, it is not as exciting and general as it could be.

Reviewer 2



Although the paper is overly technical, it addresses an important problem in deep learning with graph data. In terms of results, the novelty of the paper is in proving the universality for the equivariant case in a limited setting; this limitation to the setting where the output is a function over “nodes of the (hyper) graph” should be clarified in the abstract. For the invariant case, the advantage of the proposed technique over the proof of Maron et al’19 is that it shows that uniform approximation is possible to a function over graphs of varying size. The disadvantage is that it provides no bound on the order of equivariant tensors produced in the hidden layers. I have not checked the proofs in the appendix and cannot comment on the significance of the techniques. However, I wonder if the authors can comment on the viability of similar techniques to show the universality of equivariant networks for a broader class of discrete group actions (e.g., when the feed-forward layer is uniquely equivariant to a group action)?

Reviewer 3



This paper proves the universality of invariant and equivariant graph neural networks by using the Stone-Weierstrass theorem. This paper is original. The property of invariance and equivariance of graph neural networks has seldom been formalized in the existing literature. This paper lays a solid theoretical foundation on the universality of invariant and equivariant graph neural networks defined by a single set of parameters. However, unless the graph neural network proposed in Equation 1 covers a large number of existing graph neural networks, it is needed to validate the superiority of the proposed graph neural networks over other graph neural networks with empirical studies such as graph classification. Due to lack of generalization ability to existing graph neural networks or experimental studies, at this stage the contribution of this paper is of weak significance to the society of graph deep learning. Other aspects of this paper are summarized below: Quality The quality of this paper largely depends on the correctness on proof of the proposed theorems. I haven't check all the proof carefully. Clarity This paper is well written with clear language. There are occasional circumstances that some notations are not explained when they appear, for example: * Page 2, line 86: what does ‘i1,…id’ represents? * Page 3, line 90: what does [n]={1,…,n} mean? * Page 3, line 115, Equation 1, what is b?

[Author Response · NeurIPS 2019]

We kindly thank the reviewers for their detailed reviews, valuable feedback and suggestions for improvement. We try to
answer their questions and concerns as best as possible below.

**Reviewers 1/2/3: Arbitrary permutation group, high-order outputs.**   A common concern from all reviewers is
that the adaptation of our results to arbitrary permutation (sub)groups should be discussed more. We agree and will
provide more discussion and clarification in the final paper. In the *invariant* case, our Stone-Weierstrass (SW) approach
can actually be easily adapted to handle arbitrary subgroup of the full permutation group (since the separation of points
is still valid, etc.), we will explain this. However this extension is not valid in the *equivariant case*, which is our main
result. Indeed, our proof of the new SW theorem relies on an "ordering" of the coordinates of arbitrary equivariant
functions, and therefore requires the full permutation group to be available. We agree that a fully general version of the
SW theorem under arbitrary finite group action would be desirable, however the proof is out of our reach as of today.
Nevertheless, we believe that the proposed theorem, and its application to graph neural nets (in which full permutation
is natural), is an important first step toward a more general theory, especially since many practical applications of GNNs
consider the equivariant case. We emphasize again that equivariant universality is, in our point of view, fundamentally
more difficult than the invariant case: indeed, all the classical algebraic tools such as the powerful theory of invariant
polynomials used by [3] or the classical SW theorem do not apply, and new tools need to be developed from scratch.

The reviewers also ask for clarifications over the possibility of high-order *outputs* in the equivariant case, as opposed to
a signal over the nodes. It is indeed another limitation of our SW theorem, for the same reason that our proof requires
an "ordering" of the output. In a way, this limitation is similar to the distinction between "point clouds" (which in
dimension one can be represented by a vector up to permutation), and graphs. In the same way that the universality proof
is significantly more involved for graphs input than for point clouds input [4], high-order outputs may be significantly
more difficult to handle than vector outputs. We will add this discussion in the paper, and mention it in the abstract.

**Reviewer 1.**   The main concern of the reviewer seems to be related to arbitrary group actions as well as high-order
outputs, please see the paragraphs above. Additionally, we would like to kindly mention again that the equivariant case
is somehow fundamentally different from the invariant one, since most algebraic tools do not exist anymore. While
we agree that at first glance the equivariant result can seem incremental with respect to the invariant one, we strove to
demonstrate the significant difference between the two by including "sketches of proof" in the paper. We will emphasize
this in the final version.

**Reviewer 2.**   The main concern of the reviewer is the applicability to a broader class of permutation groups, see
paragraph above.

**Reviewer 3.**   The main concern of the reviewer is the relation between the studied GNN and other known architectures,
and experimental comparison. As mentioned in the paper, the architecture that we study is actually not new: it is a one-
layer instantiation of the GNNs proposed by [2]. In its "deep" original version, it covers all type of "Message-Passing"
GNNs, but not spectral GNNs which use powers of the adjacency matrix. We will clarify this in the final version.

We would like to kindly emphasize again that our paper is theoretical in nature. As with Multilayer Perceptron, we
cannot expect the studied one-layer GNNs to outperform deep architectures, and do not aim to, hence the limited
experimental section for illustrative purpose only. Moreover, the original paper [2] already performs many experiments
(in the deep case), showing the practical interest of these GNNs. Nevertheless, we still believe that our results have a
theoretical interest for the community, since universality was not known in the equivariant case, which covers many
practical applications. In particular, some key ideas can be extracted from our work: for instance, the notion of
"self-separability" introduced for the equivariant SW theorem could serve to adapt the results of the recent preprint [1]
(which came out after the submission deadline) to the equivariant case.

Finally, the reviewer suggests that we extend the comparison between GNNs and group invariant NNs.: the only
difference is that we consider the full set of permutation while group invariant NNs consider any arbitrary subgroup.
This is related to the first paragraph of this document, and we will add a clarification in that direction.

# References

[1] Z. Chen, S. Villar, L. Chen, and J. Bruna. On the equivalence between graph isomorphism testing and function approximation
with GNNs. *Arxiv preprint arXiv:1905.12560*, pages 1–19, 2019.

[2] H. Maron, H. Ben-Hamu, N. Shamir, and Y. Lipman. Invariant and Equivariant Graph Networks. In *ICLR*, pages 1–13, 2019.

[3] H. Maron, E. Fetaya, N. Segol, and Y. Lipman. On the Universality of Invariant Networks. In *International Conference on
Machine Learning (ICML)*, 2019.

[4] M. Zaheer, S. Kottur, S. Ravanbakhsh, B. Poczos, R. Salakhutdinov, and A. Smola. Deep Sets. (ii):1–26, 2017.


[Meta-Review · NeurIPS 2019]

This paper uses the framework of Stone-Weiserstrass theorem to prove universal approximation of equivariant/invariant functions with respect to permutation groups by a family of single-hidden layer graph neural networks. Reviewers agreed that this work is technically sound, and offers a complementary perspective of known universal approximation results under (discrete) symmetries. Some reviewers were skeptical about the significance of this work, in the sense that it may feel incremental with respect to Maron et al.'19 which establish universal approximation in the invariant case. After discussions with reviewers and based on author feedback, ultimately the AC considers the positive aspects contributed in this work outweight its shortcomings, and therefore recommends acceptance.